# Multiple Tumors in a Patient with Interleukin-2-Inducible T-Cell Kinase Deficiency: A Case Report

**DOI:** 10.3390/ijms252313181

**Published:** 2024-12-07

**Authors:** Michela Di Filippo, Ramona Tallone, Monica Muraca, Lisa Pelanconi, Francesca Faravelli, Valeria Capra, Patrizia De Marco, Marzia Ognibene, Simona Baldassari, Paola Terranova, Virginia Livellara, Valerio Gaetano Vellone, Maurizio Miano, Loredana Amoroso, Andrea Beccaria

**Affiliations:** 1Department of Neuroscience, Rehabilitation, Ophthalmology, Genetics and Maternal-Infantile Sciences, University of Genoa, 16147 Genoa, Italy; michela.difilippo1@gmail.com (M.D.F.); lisa.pelanconi@edu.unige.it (L.P.); 2D.O.P.O. Clinic, Department of Pediatric Hematology and Oncology, IRCCS Giannina Gaslini Institute, 16147 Genoa, Italy; ramonatallone@gaslini.org (R.T.); monicamuraca@gaslini.org (M.M.); 3Genomics and Clinical Genetics, IRCCS Giannina Gaslini Institute, 16147 Genoa, Italy; francescafaravelli@gaslini.org (F.F.); valeriacapra@gaslini.org (V.C.); 4Medical Genetics Unit, IRCCS Giannina Gaslini Institute, 16147 Genoa, Italy; patriziademarco@gaslini.org (P.D.M.); marziaognibene@gaslini.org (M.O.); simonabaldassari@gaslini.org (S.B.); 5Hematology Unit, Department of Pediatric Hematology and Oncology, IRCCS Giannina Gaslini Institute, 16147 Genoa, Italy; paolaterranova@gaslini.org (P.T.); mauriziomiano@gaslini.org (M.M.); 6Oncology Unit, Department of Pediatric Hematology and Oncology, IRCCS Giannina Gaslini Institute, 16147 Genoa, Italy; virginialivellara@gaslini.org; 7Pathology Unit, IRCCS Giannina Gaslini Institute, 16147 Genoa, Italy; valeriogaetanovellone@gaslini.org; 8Department of Integrated Surgical &Diagnostic Sciences (DISC), University of Genoa, 16147 Genoa, Italy; 9Pediatric Oncology and Oncohematology Unit, Department of Maternal Infantile and Urological Sciences, Policlinico Umberto I, Sapienza University of Rome, 00161 Rome, Italy; l.amoroso@policlinicoumberto1.it

**Keywords:** inborn error of immunity (IEI), *ITK* deficiency, multiple tumors, case report, childhood cancer survivors (CCS)

## Abstract

Immune dysregulation in Inborn Errors of Immunity (IEI) shows a broad phenotype, including autoimmune disorders, benign lymphoproliferation, and malignancies, driven by an increasing number of implicated genes. Recent findings suggest that childhood cancer survivors (CCSs) may exhibit immunological abnormalities potentially linked to an underlying IEI, along with a well-known increased risk of subsequent malignancies due to prior cancer treatments. We describe a patient with two composite heterozygous pathogenic variants in the interleukin-2-inducible T-cell kinase (*ITK*) gene and a history of multiple tumors, including recurrent Epstein–Barr virus (EBV)-related nodular sclerosis and Hodgkin’s lymphoma (NSHL), associated with unresponsive multiple hand warts, immune thrombocytopenia, and an impaired immunological profile (CD4+ lymphocytopenia, memory B-cell deficiency, reduction in regulatory T-cells, and B-cell- and T-cell-activated profiles). In our case, *ITK*-related immune dysregulation and prior exposure to oncological treatments seem to have simultaneously intervened in the same individual, leading to the development of a unique clinical profile. It is essential to raise awareness of the two-way association between immune dysregulation disorders and multiple tumors.

## 1. Introduction

Inborn Errors of Immunity (IEI) are characterized by a broad-spectrum phenotype including autoimmune disorders, benign lymphoproliferation, and malignancies [1].This field turns out to be constantly expanding due to the increasing number of genes involved in its development [2]. The mechanisms leading to carcinogenesis vary depending on the type of immunological defect, particularly involving surveillance mechanisms, which are still not fully understood. In some cases, lymphomas may be the first manifestation of an underlying altered immunological background, thus raising the need to discuss the importance of genetic screening for IEI in cancer subjects [3].

In childhood cancer survivors (CCS) several immunological abnormalities are described [4,5,6,7]. Although initially these conditions were considered secondary to cancer treatment, it is now well known that, in the presence of warning signs, these abnormalities may be present due to an underlying IEI [8].

In addition, CCSs have a 5- to 10-fold higher risk of developing subsequent neoplasms (SNs) than the general population, especially when exposed to high-dose radiotherapy and when in presence of a genetic cancer predisposition [9,10,11].

Among IEIs, mutations in the interleukin-2-inducible T-cell kinase (*ITK*) gene are increasingly recognized as a cause of immune dysregulation with a heightened susceptibility to autoimmunity and malignancies. *ITK* is essential for T-cell signaling and its defects result in an impaired T-cell responses, making individuals more vulnerable to infections and cancer, particularly lymphomas [12,13,14,15].

In this case report, we describe a patient with multiple tumors including Hodgkin lymphoma, which is a giant cell tumor of soft tissues and an undifferentiated pleomorphic sarcoma accompanied by recurrent warts and chronic immune-mediated thrombocytopenia, for whom genetic investigations identified pathogenic variants in *ITK*.

## 2. Case Presentation

We present the case of a boy, who was born preterm, and is a fourth child of non-consanguineous Caucasian parents. The family history included the following several cases of oncological conditions: the mother died of metastatic liver cancer at the age of 55; the father died of lung carcinoma at the age of 62 (he was a smoker and abused alcohol); the paternal grandfather died of an unspecified cancer at age of 62; the maternal aunt died of breast cancer before the age of 50; and the maternal uncle died of bone cancer at the age of 50. It is also worth noting that the mother had 13 pregnancies, 7 of which ended in miscarriages (Figure 1).

At the age of 7, the patient presented with malnutrition and poor harmonic growth (stature and weight < 2 SDS, according to the Tanner chart, and was below the genetic target). Subsequently, the patient was diagnosed with a growth hormone deficiency and required hormone replacement therapy, leading to a good recovery of auxological parameters and reaching the lower limit of the genetic target for stature.

At the age of 13, the patient developed right lateral cervical lymph node swelling, whichextended to the supraclavicular region. A serological test for Toxoplasma gondii, Rubella virus, Cytomegalovirus, and Herpes simplex (TORCH complex) and an intradermal Mantoux test were both negative. A serological test for Epstein–Barr virus (EBV) showed positivity for IgG VCA (++) and IgG EBNA (+), but negativity for IgM VCA. Unfortunately, viral load identification through PCR was not performed as the method was not available at that time.

A biopsy was performed, revealing a histological diagnosis of nodular sclerosis Hodgkin lymphoma (NSHL), stage IIA (CD30+, CD15+, CD45-, CD20-, CD42 Ro-, CD68-, EMA-, and S100-Citokeratyn-). Therefore, he underwent treatment with chemotherapy according to the AIEOP MH89 protocol [16] (including Bleomycin 60 mg/mq, Doxorubicin 150 mg/mq, Vinblastine 36 mg/mq, and Dacarbazine 2250 mg/mq) and cervical radiotherapy (20 Gy). Four months after the end of the treatment, the patient experienced local (lateral cervical) and distant (mediastinal and subdiaphragmatic) recurrences.

Biopsy was not performed due to poor clinical conditions (pleural and pericardial effusion) and chemotherapy was administered according to the IEP regimen (Ifofosfamide, Etoposide, and Prednisone). Complete remission was consolidated with the high-dose treatment regimen with a total body irradiation (TBI) (12 Gy), Cyclophosphamide (120 mg/m^2^), and continuous infusion of Vincristine (4 mg/m^2^), followed by autologous hematopoietic stem cell transplantation (HSCT).

At the age of 17, the patient presented with isolated recurrent chronic immune thrombocytopenia (with unknown anti-platelet antibody levels), which was treated discontinuously with immunoglobulins and steroids (recovery was achieved after 8 years). He also developed recurrent common hand warts, which were unresponsive to local therapy (cryotherapy, potassium hydroxide 10% solution, and laser therapy). The patient exhibited poor patient treatment compliance and refused further treatment.

At the age of 20, due to the appearance of painful subclavicular nodular swelling of about 5 cm on the same side previously irradiated, an incisional biopsy was performed. Histology was suggestive for the diagnosis of giant cell tumors of soft tissues (CD68+, S100-, and CD1a-), which were later confirmed by excision. Due to the non-radical nature of the procedure, additional local radiotherapy (58 Gy) was required.

At the age of 29, a painful muscle–tendon swelling occurred in the same site as the previous tumor. A targeted needle biopsy led to a diagnosis of undifferentiated pleomorphic sarcoma (Caldesmone+, smooth muscle actin+/, S100-, CD34-, Desmin-, CK AE1-AE3-, CKMNF116-, and EMA-). Treatment involved a personalized protocol accompanied by chemotherapy (Ifosfamide 3000mg/mq/day and Epirubicin 30 mg/mq/day for six cycles) and subsequent radical surgery (resection of the first, second, and third right ribs, resection of T1 and T2 nerve roots of the brachial plexus, and atypical resection of the upper lobe of the lung infiltrated by the tumor).

At the age of 31, at follow-up oncology check-ups, an asymptomatic mucoepidermoid carcinoma of the parotid gland was diagnosed and surgically removed. At the age of 39, during cutaneous screening for past radiation therapies, the following lesions were found: two nodular basal cell carcinomas, one on the left hip and the other on the right shoulder, both treated with complete excision; and a giant pendulous fibroma located on the left side of the waist, not yet removed. Lastly, a probable unilateral adrenal myelolipoma was diagnosed incidentally, currently under radiological follow-up. The clinical oncological timeline of the patient is described in Figure 2.

Over time, he presented with several late effects [17]: metabolic syndrome according to American Heart Association/National Heart, Lung, and Blood Institute criteria (HDL < 50 mg/dL, triglycerides > 150 mg/dL, and fasting glucose > 110 mg/dL) [18]; non-alcoholic fatty liver disease (NAFLD); bilateral cataracts; post-radiation cerebral cavernomas identified at the age of 28; iatrogenic hypothyroidism treated with a hormone replacement therapy; moderate-to-severe restrictive lung disease treated with an inhaled steroid therapy; and dysplastic nevi (>50).

Furthermore, due to previous exposure to radiation therapy, he recently underwent intestinal cancer screening with fecal immunochemical tests, which were positive for occult blood. Abdominal ultrasonography showed a suspicious polypoid image at the gastric level. An endoscopy was suggested but the patient did not accept the procedure.

Given the history of multiple tumors, a thorough genetic consultation was performed, and the following diagnostic investigations were conducted. At the age of 30, a karyotype and a low-resolution array-CGH (60 Kb) were performed, both with negative results. In the same year, the patient underwent a TP53 mutational genetic test, but no pathogenetic variant was identified. Due to the complexity of the case and considering the genetic analysis technologies that are currently available, at the age of 39, a high-resolution array-CGH (180 Kb) was conducted, which did not reveal significant variations. Finally, the genetic investigation was completed with the whole-exome sequencing (WES), which identified two heterozygous pathogenic variants in the *ITK* gene, i.e., a missense variant c.1003C>T (p.R335W) located in exon 11 and a frameshift variant c.1664delG (p.G555Afs*37) in exon 16 (according to the canonical transcript NM_005546.4). RNA isolation from peripheral blood, RT-PCR amplification of the *ITK* coding region encompassing exons 11–16, cloning into bacterial plasmids, and sequencing demonstrated that the variants occurred in trans, according to an autosomal recessive inheritance (Figure 3).

A heterozygous pathogenic variant c.8395_8404del (p.F2799Kfs*4) was also identified in the ataxia telangiectasia mutated (*ATM*) gene.

Furthermore, due to the persistence of warts unresponsive to topical therapy (Figure 4), an immunological profile analysis was performed.

Immunoglobulin levels and autoimmune screening (thyroid, celiac disease, anti-nuclear antibodies, and anti-extractable nuclear antigens) were normal while vaccine titers showed noncoverage for diphtheria, tetanus, and hepatitis B. The lymphocyte subpopulation revealed a moderate reduction in CD4+ cells (based on reference values) [19], a decrease in regulatory T-cells and memory B-cells, and an increase in activated lymphocytes (CD3+HLADR+). The study of T-cell maturation showed a shift toward the effector memory compartment with a significant reduction in naive lymphocytes (especially T CD4+ cells) and an increase in suppressor terminally differentiated effector cells (TEMRA). Additionally, the extended analysis of the B lymphocyte profile identified a notable increase in double-negative B-cells (DNB) and double-negative B-2 (DNB-2) subpopulation, along with an increase in CD21lowCD38low cells (Table 1). Conversely, an increase in natural killer (NK) cells with normal natural killer T (NKT) values was observed. CD3+TCRalpha/beta+CD4-CD8- (double-negative T-cells (DNT)) cells’ values were in normal range and the study of FAS-mediated apoptosis was negative, while a significant increase in cytokine levels (FASL, IL-10, and IL-18) was documented.

Given the identified pathogenic variants of *ITK* and the documented association between *ITK* disorders and EBV-related lymphomas [16], the Epstein–Barr encoding region (EBER) in situ hybridization was performed on a paraffin-embedded section of lymphoma in its onset (Figure 5). The analysis confirmed widespread positivity for EBV.

At the last follow-up visit, the evaluation of EBV copies in peripheral blood by using a PCR technique yielded positive results, with 1183 EBV-DNA copies per 100.000 lympho-monocytes extracted from 200 µL of whole blood.

Currently, the patient is undergoing an active follow-up for his previous oncological pathologies and is being monitored for potential late effects at the Diagnosis Observation Prevention After Oncological Therapy (D.O.P.O.) Clinic of the Giannina Gaslini Institute. He is in a good overall clinical condition, had good control over the recently developed late effects, and has no clinical signs of lymphoproliferation.

## 3. Discussion

To the best of our knowledge, this is the first report on EBV-related lymphoma recurrence and multiple tumors associated with *ITK*-related immune dysregulation.

The *ITK* is a non-receptor protein tyrosine kinase expressed in T-cells, NK cells, NKT cells, and mast cells. It plays a key role in TCR signaling, T-cell differentiation, IL-2 production, and the maturation of NKT cells, resulting in supported antiviral and antitumor properties of immunity [12]. Pathogenic homozygous variants in the *ITK* gene can give way to a wide spectrum of different phenotypic expressions, including isolated hypogammaglobulinemia and varying degrees of T-cell loss with low counts of naive CD4+ cells and NKT cells, susceptibility to recurrent infections, autoimmune phenomena, autoimmune lymphoproliferative syndrome with an increased risk of EBV-related Hodgkin and non-Hodgkin lymphoma, hemophagocytic lymphohistiocytosis (HLH), and atypical epidermodysplasia verruciformis with a predisposition to human papilloma virus (HPV) skin infection [12,13,14,15].

Our patient presents a clinical and immunological profile partially in line with what is described.

The patient has an EBV-related lymphoma, confirmed by EBER in situ hybridization, which strengthens the hypothesis that the identified *ITK* deficiency predisposes the development of neoplasia, as reported in the literature [8,12,20].

We also observed chronic immune-mediated thrombocytopenia and multiple warts, indicative of an atypical epidermodysplasia verruciformis, as manifestations of the underlying immune dysregulation. Immunologically, the patient showed a reduction in CD4+ T-cells, with a marked shift in the T-cell compartment from naïve (significantly reduced) to memory/effector cells, and a notable increase in suppressor TEMRA. Additionally, there was a general rise in activated T-cells and a reduction in regulatory T-cells. It is well known that *ITK*-deficient patients often show an increased memory–effector T-cell phenotype, indicating altered differentiation within the conventional T-cell lineage. This condition can lead to progressive CD4+ T-cell lymphopenia, elevated susceptibility to EBV, and EBV-driven lymphoproliferative diseases [12,21,22].

NK cell counts showed an increase, while NKT lymphocyte levels stayed within the normal range. This lymphocyte distribution is unusual given that *ITK* deficiency typically results in a reduced number of NKT cells. *ITK* deficiency can impair NK cell function by disrupting these pathways; however, this does not necessarily correlate with an increase in NK cell numbers [12,22]. We hypothesize that the increase in NK cells may reflect their involvement in controlling persistent low-level Epstein–Barr virus (EBV) infections [23] and concurrent persistent warts [24,25].

Moreover, regarding the B-cell compartment, the patient had reduced levels of memory B-cells and showed no response to vaccination; he did not present with hypogammaglobulinemia. This was accompanied by a notable increase in CD21lowCD38low and DNB/DNB-2 lymphocytes. This pattern, combined with the observed T-cell profile, suggests that chronic lymphocyte activation is likely exacerbated by persistent viral stimulations [26,27]. These findings imply an underlying immune dysregulation, resembling features seen in common variable immunodeficiency, particularly in IEI [1,28,29]. This profile may also explain the previously reported immune thrombocytopenia, although detailed immunological data at its onset are limited.

It is well known that IEI, of which *ITK* is a part of, presents a broad phenotype that includes the onset of malignant pathologies, particularly lymphomas [1,20,30,31]. Lymphomas can be associated with underlying immune dysregulation, particularly T-cell defects. In fact, it can appear before the diagnosis of IEI in one out of five cases [32].

In our case, the recurrence of lymphoma, family oncological history, immune thrombocytopenia, persistent warts, peculiar T-/B-cell subset with memory B-cell deficiency, reduction in regulatory T-cells, and low vaccine titers are additional factors that directed us towards a possible hereditary immune dysregulation, in line with the warning signs published by Ballow et al., proposed to assist in the diagnosis of IEI in patients with previous tumors [8].

Furthermore, a certain genetic background has been observed in hematological tumors (leukemia and lymphomas), given that 10% are genetically determined. This percentage increases with the use of new genetic diagnostic technologies, such as next-generation sequencing (NGS) and whole-exome sequencing (WES) [33]. Currently, this increased diagnostic accuracy allows for more precise identification of genetic defects in IEI associated with hematological diseases, particularly in cases with a mild phenotype diagnosed even in adulthood [29].

Regarding the subsequent neoplasms developed by our patient after suffering from lymphoma, these could be due to the progressive exposure to chemotherapy and radiotherapy [9,10]. However, given the presence, in our case, of a heterozygous pathogenic variant in the *ATM* gene, we cannot rule out an abnormal response to DNA damage related to this variant. Supporting this hypothesis, heterozygous mutants for *ATM* show intermediate radiosensitivity in vitro compared to controls and homozygous individuals [34]. This radiosensitivity, combined with exposure to harmful radiation, may trigger genomic instability and promote carcinogenic processes [35]. However, there is no conclusive evidence that carriers of a pathogenic *ATM* mutation, who have been diagnosed with cancer, have increased cancer risk secondary to radiation therapy compared with noncarriers [36,37]. Further studies to validate this hypothesis need to be conducted.

Therefore, in our case, two independent risk factors seem to have simultaneously occurred in the same individual, generating a significant oncological risk and leading to a clinical profile of multiple tumors. These factors include immune dysregulation and an increased risk of developing lymphomas due to pathogenic variants of *ITK* [16] and exposure to multiple chemo-radiotherapeutic treatments [9,10,38]. A potential additional rule of the identified *ATM* variant in cancer predisposition of this patient might also be considered. 

As suggested by Blatt et al., it is crucial to establish a follow-up and careful analysis in CCSs. This allows the identification of long-term effects related to radiotherapy and the detection of any predisposing genetic alterations, which may be shared within the same family nucleus [11]. Several cases of cancer have been reported within the patient’s family as well as multiple maternal miscarriages, which emphasize the hypothesis of a potential genetic basis. This highlights the importance of a thorough evaluation of hereditary risk factors, especially when accompanied by immune dysregulation or syndromic features [8]. Unfortunately, in our case, the initial genetic investigations did not reveal significant results. It was only thanks to modern genetic diagnostic technologies such as NGS/WES, which were not previously available, that variants in *ITK* were identified. A significant limitation of our study was the lack of genetic data on the other family members, as both parents died of cancer and the patient declined to consent to extending the analysis to his siblings. This information would have allowed a further clarification of the genetic framework and the genotype–phenotype correlations within the pedigree.

Considering these data, it is now crucial to raise awareness regarding the association between IEI and cancer in CCSs. This would allow early diagnosis of inherited immune dysregulation and of the ability to define personalized therapeutic plans.

Based on the current literature, it is suggested that every patient diagnosed with *ITK* deficiency should be considered for curative treatment by undergoing allogeneic or haploidentical HSCT at an early age [16]. For our patient, HSCT would not be a feasible and justifiable treatment, given his current state of well-being and the lack of data, to our knowledge, on adult patients and of concrete improvement in long-term outcome. Further studies on HSCT in *ITK* deficiency are warranted to define its feasibility and to investigate early and late outcomes of this treatment.

## 4. Conclusions

In conclusion, a patient with a relapsing lymphoma and a complex clinical phenotype characterized by multiple tumors, autoimmunity, and chronic infections, such as recurrent warts, should raise suspicion of an underlying immune dysregulation. This should be investigated through immunophenotyping and genetic testing, if possible, by using innovative techniques like WES.

This report aims to pioneer new approaches to diagnosing immune dysregulation in cancer survivors and to strengthen the emerging evidence linking immunodeficiency with cancer. Based on our experience, we believe it is appropriate to consider early genetic tests for IEI in patients with cancer, as well as a careful oncological screening for patients with IEI alone.

Further studies are needed to establish an appropriate screening program and to structure a multidisciplinary approach involving the geneticist.

## Figures and Tables

**Figure 1 ijms-25-13181-f001:**
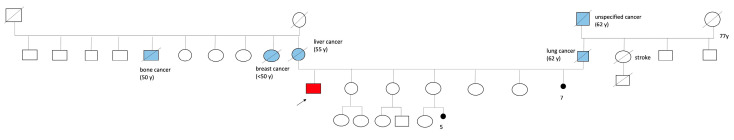
Family tree (narrow and red: patient case, blue color: relatives with a history of oncological condintions).

**Figure 2 ijms-25-13181-f002:**
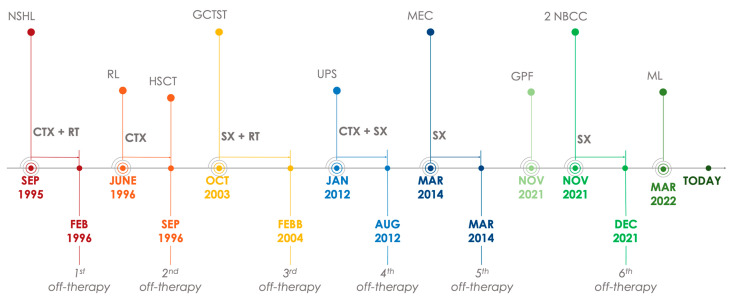
Clinical oncological timeline. Abbreviations: CTX (chemotherapy), RT (radiotherapy), SX (surgery), NSHL (nodular sclerosis Hodgkin lymphoma), RL (relapse), HSCT (hematopoietic stem cell transplantation), GCTST (giant cell tumor of soft tissues), UPS (undifferentiated pleomorphic sarcoma), MEC (mucoepidermoid carcinoma), GPF (giant pendulous fibroma), NBCC (nodular basal cell carcinoma), ML (adrenal myelolipoma).

**Figure 3 ijms-25-13181-f003:**
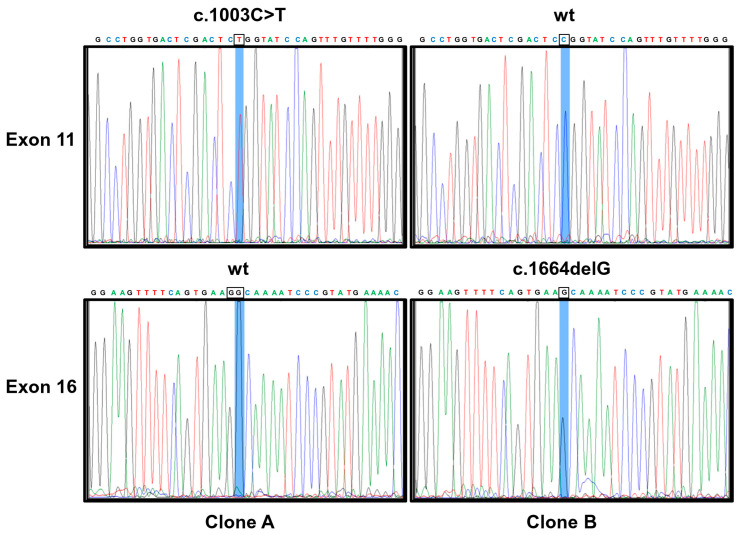
Sanger sequencing electropherograms of plasmids (Clone A and Clone B) containing *ITK* PCR products encompassing exons 11–16 (824 bp) showing that two mutations, c.1003C>T and c.1664delG, are in trans since that the plasmid containing the allele harboring one mutation is wild-type (wt) for the second one. Methods: Total RNA was isolated from whole blood using PAXgene blood RNA tubes (Quiagen, Hilden, Germany) using the nucleic acid purification kit (PAXgene Blood RNA Kit, Quiagen, Hilden, Germany) according to the standard procedures. Reverse transcription to cDNA was carried out in a 20μL reaction mixture with the use of SuperScript II Reverse Transcriptase (Invitrogen, Carlsbad, CA, USA). Moreover, 10 μL of synthesized cDNA was used to amplify the *ITK* coding region encompassing exons 11–16, which include both mutations (the *ITK* c.1003C>T and *ITK* c.1664delG) with the following primers: ITK-10F: ATCACCAACATAATGGAGGAG; ITK-16R: CTCTTTCCAGCAGTGATTC.RT-PCR products were analyzed by electrophoresis on 2% agarose gels and viewed by ethidium bromide staining. RT-PCR products isolated by agarose gel were purified and cloned into the TA cloning vector pGEM-T (Promega, Madison, WI, USA) to transform *E. coli* JM109-competent cells. Transformants were picked up and grown over night. Plasmid DNA was extracted with a Quiagen Plasmid Mini kit (Quiagen, Hilden, Germany) and sequenced using specific *ITK* primers by utilizing the BigDye Terminator v 3.1 Cycle Sequencing kit (Thermo Fisher Scientific, Waltham, MA, USA).

**Figure 4 ijms-25-13181-f004:**
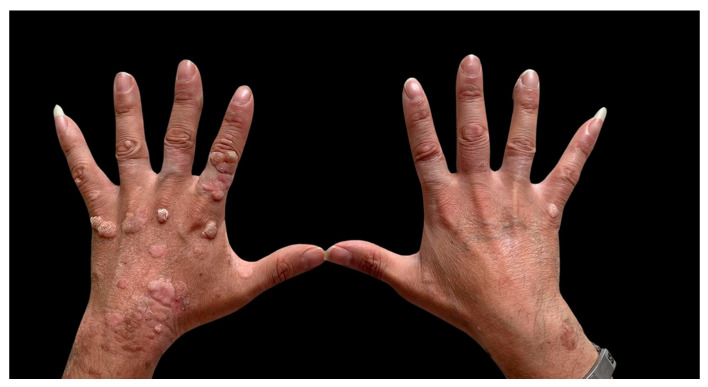
Hand warts of the patient.

**Figure 5 ijms-25-13181-f005:**
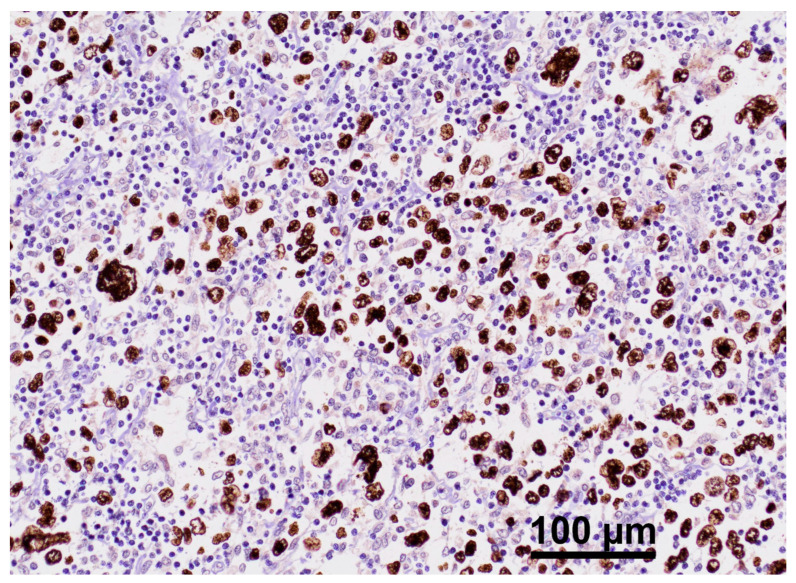
EBER ISH stain of lymphoma tissue (original magnification of 200×). This image shows an EBER in situ hybridization (ISH) stain of a lymphoma tissue sample. Neoplastic cells appear stained in brown, indicating positivity for Epstein–Barr virus-encoded RNA (EBER). This suggests the presence of EBV infection in these cells. Surrounding lymphatic cell results are negative for EBER.

**Table 1 ijms-25-13181-t001:** Peripheral immunological profile of the patient.

Parameter	Values	Reference Range
Total lymphocytes (/mmc)	1570	1080–3280
CD3+CD4+ (helper T-cells) (/mmc)	390↓	460–1232
CD3+CD8+ (suppressor T-cells) (/mmc)	408	187–844
CD19+ (B-cells) (/mmc)	193	92–420
CD3-CD56+CD16+ (NK cells) (/mmc)	495↑	89–322
CD3+CD56+CD16+ (NKT cells) (% total lymphocytes)	4.4	3.3–4.6
CD3+CD4+CD45RA+ CD27+ (naïve TH)(% CD4+)	0.6↓	49–72
CD3+CD4+CD45RA-CD27+ (TH central memory) (% CD4+)	52.7↑	24–43
CD3+CD4+CD45RA-CD27- (TH effector memory) (% CD4+)	46.2↑	2–7.5
CD3+CD4+CD45RA+CD27- (TH terminally differentiated effector cells) (% CD4+)	0.3↓	2–7.5
CD3+CD8+CD45RA+ CD27+ (naïve TS) (% CD8+)	12.6↓	48.5–87.5
CD3+CD8+CD45RA-CD27+ (TH central memory) (% CD8+)	15.1	9.5–38
CD3+CD8+CD45RA-CD27- (TH effector memory) (% CD8+)	8.7↑	0.2–7.0
CD3+CD8+CD45RA+CD27- (TH terminally differentiated effector cells) (% CD8+)	63.4↑	0.8–14
CD3CD4CD25brCD45RA- (regulatory T-cells) (% total lymphocytes)	0.2↓	0.6–0.8
CD3+TCR γδ+ (% total lymphocytes)	4.5	3.3–4.6
CD3+HLA DR+ (% total lymphocytes)	23.9↑	4–6
CD3+ TCR αβ+CD4-CD8- (double-negative T) (% total lymphocytes)	1.4	<1.7
CD3+CD25+/CD3+HLADR+ ratio	0.2↓	>1
CD19+CD27+ (memory B-cells) (% total lymphocytes)	12.5↓	>15
CD19+CD27-CD10+-CD38++ (transitional B-cells) (% B-cells)	1.7↓	2.0–6.0
CD19+CD27-CD10+-CD38+IgD+ (naïve B-cells) (% B-cells)	79.1↑	58–75
CD19+CD27+IgD+IgM+ (marginal-zone B-cells) (% B-cells)	4.9↓	7.5–21.5
CD19+CD27+IgD-IgM- (switched memory B-cells) (% B-cells)	8.0	7.0–19.0
CD19+CD21lowCD38low (% B-cells)	7.5↑	<5.0
CD19+CD27-IgD- (double-negative B)(% B-cells)	16.1↑	<10.0
CD19+CD27-IgD-CD21low (double negative B2)(% B-cells)	41.3↑	<25.0
Immunoglobulin A (mg/dL)	164	70–400
Immunoglobulin G (mg/dL)	1252	700–1600
IgG1 (mg/dL)	928	490–1140
IgG2 (mg/dL)	181	150–640
IgG3 (mg/dL)	103	20–110
IgG4 (mg/dL)	84	8–140
Immunoglobulin M (mg/dL)	67	40–230
FAS-Ligand (FASL) (pg/mL)	358↑	<200
Interleukin 10 (IL-10) (pg/mL)	174↑	>20
Interleukin18 (IL-18) (pg/mL)	250	>500
Vitamin B12 (pg/mL)	565	191–663
% Surviving T-cells after the FAS antigen stimulation (defective lymphocyte apoptosis) (%)	29	<78

## Data Availability

In this study, no new data were created or analyzed. Data sharing is not applicable to this article.

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
