# Peer review of "Multiple Tumors in a Patient with Interleukin-2-Inducible T-Cell Kinase Deficiency: A Case Report"

_ijms, 2024, doi:10.3390/ijms252313181_

Round 1
Reviewer 1 Report
Comments and Suggestions for Authors
‘Multiple tumors in a patient with ITK deficiency: a case report’ present case of a child with multiple tumors and immune dysregulation related to two composite heterozygous pathogenic variants in ITK.
Manuscript is written clearly and in a well-structured manner. It is relevant for the field and fits to the aim of the IJMS and special issue ‘New Insights into Immune Dysregulation Disorders’. Cited references are mostly recent publications (within the last 5 years) and relevant. Ethics statements and data availability statements are adequate.
Manuscript is scientifically sound. Manuscript’s results are reproducible based on the details given in the methods section. Figures and table are appropriate and they properly show the data. Data are easy to interpret and understand.
My major comment refers to the experimental design appropriate to test the hypothesis. This report supports the use of early genetic testing for IEI in patients with cancer based on the results of one member of the family with multiple tumors whose parents and grandparents also died of cancer at not so late age. Described patient has two composite heterozygous pathogenic variants in ITK and we may say conditionally that this confirms the hypothesis. I think that in order to provide the evidence for the arguments presented, it would be great if authors could confirm these ITK mutations in his parents too. Since their samples are not available, it would be interesting to have samples of his five sisters tested for ITK mutations and also to provide some medical data regarding their health. So please add some data about his closest siblings in order to have conclusions consisted with the evidence and arguments presented.
Minor comments: there are typo errors throughout the text that should be corrected.
Author Response
Thank you for taking the time to thoroughly review this manuscript. Below, we provide detailed responses to your comments, with the corresponding revisions clearly highlighted in the resubmitted documents. Following the request from one of the three reviewers to improve the introduction of the manuscript, the order of the references has been modified (highlighted in red) from number 12 to number 19.
Comments 1: Multiple tumors in a patient with ITK deficiency: a case report’ present case of a child with multiple tumors and immune dysregulation related to two composite heterozygous pathogenic variants in ITK. Manuscript is written clearly and in a well-structured manner. It is relevant for the field and fits to the aim of the IJMS and special issue ‘New Insights into Immune Dysregulation Disorders’. Cited references are mostly recent publications (within the last 5 years) and relevant. Ethics statements and data availability statements are adequate. Manuscript is scientifically sound. Manuscript’s results are reproducible based on the details given in the methods section. Figures and table are appropriate and they properly show the data. Data are easy to interpret and understand. My major comment refers to the experimental design appropriate to test the hypothesis. This report supports the use of early genetic testing for IEI in patients with cancer based on the results of one member of the family with multiple tumors whose parents and grandparents also died of cancer at not so late age. Described patient has two composite heterozygous pathogenic variants in ITK and we may say conditionally that this confirms the hypothesis. I think that in order to provide the evidence for the arguments presented, it would be great if authors could confirm these ITK mutations in his parents too. Since their samples are not available, it would be interesting to have samples of his five sisters tested for ITK mutations and also to provide some medical data regarding their health. So please add some data about his closest siblings in order to have conclusions consisted with the evidence and arguments presented.
Response 1: We sincerely thank the reviewer for their valuable comments and for highlighting the importance of investigating the patient’s siblings. We had indeed considered extending our study to include the patient’s sisters to evaluate the potential presence of pathogenic variants. Accordingly, we informed the patient about the significance of this investigation. However, the patient ultimately did not provide consent for this analysis. We fully recognize the importance of this investigation and have highlighted in the manuscript the significant limitation posed by the lack of genetic data from other family members (see page 9, with the added sentence highlighted in red: “and the patient declined to consent to extending the analysis to his siblings”). If the reviewer finds it helpful, we would be happy to include further consideration in the manuscript regarding the lack of consent for extending the analysis.
Comments 2: Minor comments: there are typo errors throughout the text that should be corrected.
Response 2: We agree with the reviewer’s observations and apologize for the errors. Accordingly, we have revised the text.
Addidional clarification: Please note that a modification has been made to the affiliation of three authors, highlighted in red on the first page. This change was made in order to specify the authors' respective units of affiliation . As a result, two new affiliations have been added, and the order of affiliations has been adjusted from number 5 to number 9: 5Hematology Unit, Department of Pediatric Hematology and Oncology, IRCCS Giannina Gaslini Institute, Genoa, 6Oncology Unit, Department of Pediatric Hematology and Oncology, IRCCS Giannina Gaslini Institute, Genoa.
Reviewer 2 Report
Comments and Suggestions for Authors
-This is an invaluable case report. The work is informative and is a great contribution our knowledge. A few suggestions for the authors:
-Define all abbreviations at first mention (e.g., ATK in abstract) and never use the full name again.
-The Discussion could be improved by covering ITK function and related diseases.
-The family history is a very important risk factor. Please elaborate on this aspect in the Discussion.
-So many linguistic errors (e.g., sierology, potencially... etc.).
Author Response
Thank you for dedicating your valuable time to reviewing this manuscript. Please find below detailed responses to your comments, along with the corresponding revisions, which have been highlighted in the resubmitted files. Following the request from one of the three reviewers to improve the introduction of the manuscript, the order of the references has been modified (highlighted in red) from number 12 to number 19."
Comments 1: This is an invaluable case report. The work is informative and is a great contribution our knowledge. A few suggestions for the authors: -Define all abbreviations at first mention (e.g., ATK in abstract) and never use the full name again.
Response 1: We sincerely thank the reviewer for their valuable comments. We thank you for pointing this out and we proceeded to modify the expression in the text (highlighted in red).
Comments 2: The Discussion could be improved by covering ITK function and related diseases.
Response 2: We appreciate the reviewer’s suggestion and have incorporated a sentence highlighted in red on page 7 (Chapter 3: Discussion): “The ITK is a non-receptor protein tyrosine kinase expressed in T, NK and NKT cells, and mast cells. It plays a key role in TCR signaling, T-cell differentiation, IL-2 production and the maturation of NKT cells, supporting antiviral and antitumor immunity[12].”
Comments 3: The family history is a very important risk factor. Please elaborate on this aspect in the Discussion.
Response 3: We would like to thank the reviewer for drawing our attention to this precision. We have add a sentence in the text in the discussion highlighted in red (page 9): “Several instances of cancer have been reported within the patient's family, as well as multiple maternal miscarriages, which emphasizes the hypothesis of a potential genetic basis. This underscores the importance of a thorough evaluation of hereditary risk factors, especially when accompanied by immune dysregulation or syndromic features[8].”
Comments 4: So many linguistic errors (e.g., sierology, potencially... etc.).
Response 4: We agree with this comment. Therefore, we have revised the text.
Additional clarifications: Please note that a modification has been made to the affiliation of three authors, highlighted in red on the first page. This change was made in order to specify the authors' respective units of affiliation . As a result, two new affiliations have been added, and the order of affiliations has been adjusted from number 5 to number 9: 5Hematology Unit, Department of Pediatric Hematology and Oncology, IRCCS Giannina Gaslini Institute, Genoa, 6Oncology Unit, Department of Pediatric Hematology and Oncology, IRCCS Giannina Gaslini Institute, Genoa.
Reviewer 3 Report
Comments and Suggestions for Authors
Dear Authors,
congratulations on your valuable work. Please, find here below some suggestions, that, in my opinion, will further help improving the quality of your paper.
1. Please, be ware of the typos, there are many of them. I suggest a careful revision of the manuscript.
2. Figure 1 and figure 3 are difficult to read, please improve the quality.
3. In order to guarantee patients' privacy, I suggest to edit Figure 4 removing tattoos on the left arm with an image editing tool.
4. For the correct interpretation of the case, I think it would be a good idea to include, before the case presentation, a quick literature search that better contextualises the case and your treatment choices. In this sense, I would include the search strategy and a summary of the most impactful evidence, perhaps presenting it through PICO.
5. Just one last curiosity: why were no direct treatment options against EBV considered? Were there viable options?
Author Response
Thank you for dedicating your valuable time to reviewing this manuscript. Below, you will find detailed responses to your comments, with the corresponding revisions clearly highlighted in the resubmitted files. Following the request from the reviewer to improve the introduction of the manuscript, the order of the references has been modified (highlighted in red) from number 12 to number 19."
Comments 1: Dear Authors, congratulations on your valuable work. Please, find here below some suggestions, that, in my opinion, will further help improving the quality of your paper. Please, be ware of the typos, there are many of them. I suggest a careful revision of the manuscript.
Response 1: We sincerely thank the reviewer for their valuable comments. We agree with the reviewer’s observations; the text has been revised accordingly.
Comments 2: Figure 1 and figure 3 are difficult to read, please improve the quality.
Response 2: We have enhanced the quality of the figures to the best of our ability with the assistance of our Institute's service. We hope that the figures now meet the required standards.
Comments 3: In order to guarantee patients' privacy, I suggest to edit Figure 4 removing tattoos on the left arm with an image editing tool.
Response 3: We have enhanced the quality of Figure 4 by cropping the image, as removing the tattoos would have compromised its quality. We hope the figure now meets your requirements.
Comments 4: For the correct interpretation of the case, I think it would be a good idea to include, before the case presentation, a quick literature search that better contextualises the case and your treatment choices. In this sense, I would include the search strategy and a summary of the most impactful evidence, perhaps presenting it through PICO.
Response 4: Thank you for your valuable feedback and advice. We have enhanced the introduction by better contextualizing it in relation to the clinical case presented. Please see the highlighted sentences added to the introduction (page 2): “Among IEIs, mutations in the Interleukin-2-inducible T-cell kinase (ITK) gene are increasingly recognized as a cause of immune dysregulation with a heightened susceptibility to autoimmunity and malignancies. ITK in essential for T-cell signaling and its defects result in impaired T cell responses, making individuals more vulnerable to infections and cancer, particularly lymphomas[12-15 ]. In this case report, we present a patient with multiple tumors including Hodgkin Lymphoma, a giant cell tumor of soft tissues, and undifferentiated pleomorphic sarcoma, accompanied by recurrent warts and chronic immune-mediated thrombocytopenia, for whom genetic investigations identified pathogenic variants in ITK.” Due to this addition to the text, the order of the references (from number 12 to 19) has been modified and highlighted in red.
Comments 5: Just one last curiosity: why were no direct treatment options against EBV considered? Were there viable options?
Response 5: Thank you for pointing this out. At the time of the initial tumor onset in 1995, a correlation with EBV infection had not yet been established through serology, and PCR-based methods for detecting blood viremia were not available. The diagnosis of EBV-related lymphoma was made recently, through a retrospective re-evaluation of the histological findings from the initial tumor. This assessment was further supported by the identification of the ITK mutations and the well-documented association between ITK deficiency and EBV-related lymphomas.
Additional clarifications: Please note that a modification has been made to the affiliation of three authors, highlighted in red on the first page. This change was made in order to specify the authors' respective units of affiliation . As a result, two new affiliations have been added, and the order of affiliations has been adjusted from number 5 to number 9: 5Hematology Unit, Department of Pediatric Hematology and Oncology, IRCCS Giannina Gaslini Institute, Genoa, 6Oncology Unit, Department of Pediatric Hematology and Oncology, IRCCS Giannina Gaslini Institute, Genoa.
Round 2
Reviewer 2 Report
Comments and Suggestions for Authors
The authors have adequately addressed my comments.